Characteristic changes in malt, wort, and beer produced from different Nigerian rice varieties as influenced by varying malting conditions

http://orcid.org/0000-0002-0835-5872 Ofoedu Chigozie E. 1 chigozie.ofoedu@futo.edu.ng
Akosim Chibugo Q. 1
Iwouno Jude O. 1
Obi Chioma D. 2
http://orcid.org/0000-0001-5804-7950 Shorstkii Ivan 3
http://orcid.org/0000-0003-4475-8887 Okpala Charles Odilichukwu R. 4 charlesokpala@gmail.com
1 Department of Food Science and Technology, School of Engineering and Engineering Technology, Federal University of Technology, Owerri , Owerri, Imo , Nigeria
2 Department of Food Science and Technology, Nnamdi Azikiwe University , Awka, Anambra , Nigeria
3 Department of Technological Equipment and Life-Support Systems, Kuban State Technological University , Krasnodar , Russian Federation
4 Department of Functional Food Products Development, Faculty of Biotechnology and Food Science, Wroclaw University of Environmental and Life Sciences , Wroclaw , Poland
Urban Pawel
Electronic publication date: 2021 Mar 19
Publication date: 2021
Volume: 9
Electronic Location ID: e10968
Received 2020 Oct 28; Accepted 2021 Jan 28
Copyright: © 2021 Ofoedu et al.
Copyright year: 2021
Copyright holder: Ofoedu et al.
License: This is an open access article distributed under the terms of the Creative Commons Attribution License, which permits unrestricted use, distribution, reproduction and adaptation in any medium and for any purpose provided that it is properly attributed. For attribution, the original author(s), title, publication source (PeerJ) and either DOI or URL of the article must be cited.
License URL: https://creativecommons.org/licenses/by/4.0/

Keywords: Malting conditions, Rice malt, Rice beer, Rice wort, Mashing process

Funding: Wrocław University of Environmental and Life Sciences UPWR 2.0 European Social Fund under the Operational Program Knowledge Education Development POWR.03.05.00-00-Z062/18 Publication was financed by the project UPWR 2.0: international and interdisciplinary programme of development of Wrocław University of Environmental and Life Sciences, co-financed by the European Social Fund under the Operational Program Knowledge Education Development, under contract No. POWR.03.05.00-00-Z062/18 of 4 June 2019. The funders had no role in study design, data collection and analysis, decision to publish, or preparation of the manuscript.

==============================
Gluten-free beer could be produced with rice, although the latter would primarily serve as adjunct in combination with barley malt in today’s brewing. However, the recent growing realisation of the potential and applications of rice malt for brewing an all-rice malt beer through varying malting conditions cannot be overlooked. In this study, therefore, the characteristic changes in malt, wort, and beer from different Nigerian rice varieties (FARO 44, FARO 57, NERICA 7) as influenced by varying malting conditions (steeping duration (18, 24 and 30 h), germination periods (2, 3 and 4 days) and kilning temperatures (50 and 55 °C)), were investigated. Rice (grain) samples were examined by thousand kernel weight (TKW), germinative energy (GE), germinative capacity (GC), and degree of steeping (DoS). To ensure that rice wort/beer with unique beer style and enhanced attributes, comparable to barley wort/beer is produced, malting conditions that produced rice malts with peak diastatic power (DP), cold water extract (CWE), and hot water extract (HWE) were selected. Peak DP, CWE and HWE were obtained at FARO 44 (18 h steeping, 3 days germination, 55 °C kilning (S18G3K55°)), FARO 57 (30 h steeping, 2 days germination, 50 °C kilning (S30G2K50°)) and NERICA 7 (24 h steeping, 3 days germination, 55 °C kilning (S24G3K55°)). Selected malts were further tested for moisture content, total nitrogen, malt yield and malting loss and subsequently progressed to wort and beer production. Wort’s pH, total soluble nitrogen (TSN), brix, kolbach index (KI), free amino nitrogen (FAN), dextrose equivalent (DE), original extract (OE) and sugar profile were determined, as well as beer’s pH, colour, apparent extract (AE), alcohol by volume (%ABV), turbidity and sensory attributes. Rice grain varied significantly (p < 0.05) in TKW, GE, GC and DoS across varieties. Despite wort’s pH, TSN, DE, OE as well as beer pH, colour, AE and turbidity resembling (p > 0.05) across varieties, wort’s brix, KI, FAN, sugar profile as well as beer’s %ABV, differed significantly (p < 0.05). Sensory attributes of appearance, colour, mouthfeel, and overall acceptability in beer differed noticeably (p < 0.05), except for aroma and taste (p > 0.05). Overall, the rice beer, though very slightly hazy, represented a pale yellow light lager, which is indicative of its peculiar beer style. Besides increased DP and enhanced hydrolysis, varying malting conditions of current study could serve as a pathway of reducing the cost of exogenous (commercial) enzymes or barley malt imports, together with decreasing barley’s dependency for brewing in the tropics.

Introduction

The global production of rice (Oryza sativa), to meet up with the increasing consumer demand, is projected to potentially double by 2050 (USDA–ERS, 2019; Organization for Economic Cooperation & Development/Food & Agriculture Organization (OECD/FAO), 2019). In Africa, Nigeria has led the rice production (Daoui, 2018) where, for instance, the 2018 rice paddy production recorded 6.81 million tonnes (World Data Atlas, 2020, accessed 8 June 2020). The recent curb of rice importation was targeted to facilitate increased local production (Russon, 2019). However, Nigeria increasingly imported barley malt until the 1988 ban (Koleoso & Olatunji, 1992), after which the focus on indigenous breweries intensified pushing the locally produced and commercially viable cereals like maize, rice, and sorghum to thrive. Besides rice, barley is another global cereal that breweries utilize (Contreras-Jimenez et al., 2018; Daneri-Castro, Svensson & Roberts, 2016). Non-temperate countries are largely limited to produce barley in commercial quantities and have no alternative but to supplement by the importation of (either malted or unmalted) grains for their breweries. Prior to the 1988 Nigeria barley malt importation ban, however, various pilot plant and commercial tests had established locally cultivated sorghum malt or grit as brewing candidates compared with existing barley malt brands (Koleoso & Olatunji, 1992). Resembling those of sorghum, malted maize brewing properties could potentially replace the barley malt (Okafor & Aniche, 1980). By assessing malting and brewing potentials, Okafor & Iwouno (1991) reported Nigerian rice varieties as a promising substitute for barley in beer production. Odibo, Nwankwo & Agu (2002) reported fermentable extracts from locally cultivated sorghum to keep for a longer time until required in the brewing process. Whilst Ogbeide (2011) showed sorghum as an adjunct to malted barley in the wort production/brewing process, Iwouno & Ojukwu (2012) showed the malting quality potential of a Nigerian locally cultivated yellow maize variety. Recently, Ofoedu, Osuji & Ojukwu (2019) reported that the sugar profile of local rice product/derivate (syrup) resembled that of barley wort.

Malting, within the brewing process, is employed particularly to prepare the brewing raw material. The next step that subsequently follows is the mashing and wort fermentation. It is important to reiterate that the malting process involves steeping, germination, and kilning. This process is very crucial in beer production because it helps to develop/prepare the inactive hydrolytic enzymes present in the raw grain (Dewar, Taylor & Berjak, 1997). Steeping enhances grain softening, increases water availability, and stimulates germination (Sripriya, Anthony & Chandra, 1997). Germination facilitates the production of hydrolytic enzymes which aids in grain transformation (Osuji, Ofoedu & Ojukwu, 2019). Kilning reduces the grain moisture, stops germination (arrest enzyme action), and enhances the production of malt color via Maillard reaction (Skendi & Papageorgiou, 2018). Mashing, however, facilitates the enzymatic degradation of polysaccharides that are present (in the malt) to simple sugars, which eventually converts to alcohol in the fermentation step of the beer manufacturing (Gupta, Abu-Ghannam & Gallaghar, 2010). Moreover, it has been understood that the low protein and fat content of rice have the potential to assure a slightly higher starch content of 80–90% (Wani et al., 2012; Omar et al., 2016; Othman & Omar, 2017) compared to barley starch content of approximately 70% (Asare et al., 2011; Zhu, 2017). This aspect of rice might suggest a higher extract yield (Narziss & Back, 2012), though with different starch structure, and composition in amylose and amylopectin, as well as a lower amylolytic activity than barley (Cela et al., 2020). Therefore, there is a need to optimise the malting and mashing conditions of rice. Moreover, rice has been reported to yield incomplete saccharified mash (wort) (Teeravivattanakit et al., 2017; Roberto et al., 2020) which could be due to its insufficient inherent starch-degrading enzymes (low diastatic power (DP)) and high gelatinization temperature (Cela et al., 2020), owed to the nature of rice starch.

Applying such processing methods as malting on rice can enhance the degradation (depolymerization) of its high molecular weight constituents (starch and proteins) to achieve lower molecular weight constituents (sugars and amino acids), which could eventually influence its composition and functionality during processing (Shumin et al., 2014). This opinion appears to concur with the findings of previous workers (Mayer et al., 2014; Usansa et al., 2011) wherein, despite the lower DP of rice compared to barley, rice can serve as a raw material candidate for brewing given its higher limit-dextrinase content compared to barley that elevates the complete saccharification of rice wort, provided the malting conditions are optimised. In addition, the high fiber content of rice assures an enhanced lautering process, attributable to its filtering capacity (Kongkaew, Usansa & Wanapu, 2012). The sufficient (structural) protein degradation either prior to or simultaneously with starch saccharification, according to Narziss & Back (2012), can expose the grain cell wall structure, thus enhancing the easy breakdown of its endosperm (Shumin et al., 2014; Kohorn, 2000). As the protein degradation appears more challenging in the malted rice compared to barley, the endogenous enzyme in the rice malt would facilitate the breakdown of rice constituents in vivo and improve the extract yield in the wort production. During the malting process, additionally, the rice would attain a higher alpha-amylase production (Ayerno & Hammond, 2000) together with a suitable beta-amylase activity (Cela et al., 2020). Besides starch and protein degradation of rice, the actions of malting conditions enhance enzyme production, which aids in vivo hydrolysis of endosperm in the rice kernel during germination and in vitro hydrolysis during mashing (Hassani, Zarnkow & Becker, 2013; Garzon, Torres & Drago, 2016). Varying the malting conditions, so as to ascertain the situation that would bring about rice malts with potentially higher DP, and fermentable extracts appears a promising remedy to the brewing challenges associated with rice.

Nigeria’s local rice varieties having evolved over the years with improved qualities like (longer) grain length, improved colour/cooking quality, etc., continues to compete with foreign ones, thrives increasingly, and spreads its distribution/reputation to the West African sub-region. Besides the increased interest to find local/indigenous raw material(s) to supplement barley in brewing, to help reduce (barley) imports, increase (local/indigenous) rice production (Index Mundi, 2020), Nigeria’s quest to attain self-sufficiency (in rice production) is not far-fetched, as have been described elsewhere (Ofoedu et al., 2020). Placing the greater emphasis on such underutilized indigenous rice varieties, particularly those perceived as undesirable due to widely accepted factors such as poor cooking quality (soft and sticky grains), poor physical attributes (poor colour, short-grain length, etc.), and poor consumer acceptability, therefore, makes (rice) product quality diversification very fitting. The growing realisation of researchers, (rice) processors, and local breweries about the potentials as well as the diverse applications of these (underutilized indigenous) rice varieties, have recently facilitated (rice) product diversification, to actualise promising products like syrups (Ofoedu et al., 2020), gluten-free beers (Cela et al., 2020), flours, and malts (Osuji, Ofoedu & Ojukwu, 2019). Also, even though rice grits have served as adjuncts in brewing, the use of rice malt as a specialty ingredient or base malt in the brewing industry, for instance, in brewing an all-rice malt beer (gluten-free rice beer) (Marconi et al., 2017), should be very promising. Additionally, the malted rice specifically harnessed from locally produced indigenous rice varieties in Nigeria appears not fully explored, particularly as a principal raw material or substrate for brewing. In the context of the (above-mentioned) discourse, therefore, this current work was specifically purposed to determine the characteristic changes in malt, wort, and beer produced from different (Nigerian) rice varieties as influenced by varying malting conditions. The indigenous rice varieties used for this study are locally available and in commercial quantities.

Materials and Methods

Schematic overview of the experimental program

The schematic overview of experimental study, showing the key/major stages from the procurement of rice samples through malting, wort production, and beer production to laboratory analyses, is shown in Fig. 1. Specifically, this current work was designed to determine the characteristic changes in malt, wort, and beer produced from different (Nigerian) rice varieties as influenced by varying malting conditions. This was done by investigating the impact of varying the experimental variables (steeping durations (18, 24 and 30 h), germination periods (2, 3 and 4 days), and kilning temperatures (50 °C and 55 °C)), on rice malt quality with reference to dependent variables (cold water extract (CWE), hot water extract (HWE), and diastatic power (DP)). Consequently, the rice malts with higher dependent variables (CWE, HWE, and DP) progressed to mashing and brewing. By comparing rice varieties, however, the effect of malting conditions on rice wort and beer, as exerted by the selected rice malts were determined using some key parameters/indices. Duplicate determinations were carried out by analysing aliquot samples from the sample population (rice malt, rice wort, and rice beer) across the rice varieties.

Figure 1 Schematic overview of the experimental program.

Procurement of chemicals, enzymes and rice grains

Procured from certified sources, all chemicals and reagents (i.e., Copper (II) sulphate pentahydrate (CuSO4⋅5H2O), Potassium sodium tartrate tetrahydrate (KNaC4H4O6⋅4H2O), Calcium hydroxide (Ca(OH)2), hydrochloric acid (HCl), boric acid (H3BO3), potassium sulphate (K2SO4), sulfuric acid (H2SO4), Trioxonitrate (V) acid (HNO3), sodium metabisulphite (Na2S2O5), Sodium hydroxide (NaOH), Ammonium hydroxide (NH4OH), Ninhydrin, Methylene blue indicator, Phenolphthalein indicator, Fehling’s solution, Anhydrous D-glucose) were of analytical grade standard.

Commercial exogenous microbial enzymes (namely: α-amylase (25 U/mL) and β-amylase (15 U/mL)) were procured from Nigerian breweries PLC (Awo-Omamma, Imo State, Nigeria). Protease (microbial) enzyme having 10 U/mL enzyme activity and isomerised hops used were procured from Department of Applied Microbiology and Brewing, Enugu State University of Science and Technology, Enugu State, Nigeria. Yeast strain (Saccharomyces pastorianus) was procured from the Nigerian Breweries PLC (Ama, Enugu State, Nigeria).

Improved rice grains (FARO 44, FARO 57, and NERICA 7) were purchased from National Cereals Research Institute, Amakama, Olokoro Umuahia, Abia State, Nigeria.

Rice grain analysis

Evaluation of rice grain

Grain quality analyses (such as thousand kernel weight (TKW), germinative energy (GE), germinative capacity (GC), degree of steeping (DoS), etc.) are important for the evaluation of suitability of rice variety for malting and brewing (Marconi et al., 2017). In this current work, TKW, GE, GC and DoS were determined.

Determination of TKW

Thousand kernel weight was determined according to the method described by Esiape (1994). Hundred (100) grains of paddy rice randomly selected from the bulk were weighed using a weighing balance. Each weight was multiplied by 10 to obtain the 1,000 kernel weight. Determinations were done in duplicate.

Determination of GC and GE

Germinative capacity measures grain viability, whereas GE measures the extent to which grain will germinate in a standardized test. Both rapid and complete germination are well-known essential features of good malt. For this current study, the GE and GC (presented in percent (%)) of rice samples were determined using the recommended method of analysis of the Institute of Brewing (IOB) (2007).

(1) Germinativeenergy(%)=NumberofviablegrainsTotalnumberofgrains×100

(2) Germinativecapacity(%)=%Germinativeenergy−%Dormancy

Herein, dormancy is rice grain’s inherent inability to germinate under optimal environmental conditions expressed as; (3) %Dormancy=NumberofunviablegrainsTotalnumberofgrains×100

Determination of DoS

Degree of steeping measures the amount of water readily absorbed by the grains. DoS expressed as percentage (%) was determined by the method described by Kunze (2005) with slight modification. One hundred grams rice kernels of predetermined moisture content (MC) were soaked in a 100 mL beaker containing 50 mL of distilled water at ambient temperature (28 °C ± 2). Steeping was done continuously until constant weights were attained and recorded. Soak waters were drained off the grains by the use of sieves. From the increase in mass, the DoS was calculated as; (4) DegreeofSteeping(%)=XWi×100

where X = Variable expressed in grams (g), showing the relationship between Wi, Wf and MC; Wi×(D+Mc)Wf

Wi = Mass of rice grain before steeping (g)

Wf = Mass of rice grain after steeping (g)

MC = Moisture content of rice grain (%)

D = Mass of water absorbed by rice grain (Wf – Wi) (g)

Malting of rice grain

After manual cleaning, the paddy rice of different varieties sorted to remove contaminants/damaged seeds, was winnowed to remove dust. Prior to malting, the rice paddy were disinfected in water containing 0.20% sodium metabisulphite. The malting process followed method of Kunze (2004) with some modifications. Briefly, rice samples were steeped in water at 20–25 °C for 18, 24, and 30 h with alternating steep cycle of 6 h wet-steep period and 30 min air rest. Grains were allowed to germinate, thereafter removed after 2, 3, and 4 days and kilned in hot air oven (Genlab, England, Model M 30 C, S/N 92B060) at temperatures between 50 °C and 55 °C for about 22–24 h. Kilned samples were manually de-rooted by rubbing off with hand, winnowed to remove the rootlets/dust, and milled to produce the rice grist.

Production of rice malt wort

The flow diagram of rice malt wort production presented in Fig. 2, which was slightly modified from previous studies (Okafor & Iwouno, 1991; Nwanekezi, Osuji & Onyeneke, 2007; Marconi et al., 2017; Ofoedu, Osuji & Ojukwu, 2019; Iwouno et al., 2019) of under modified cereal (rice) malts, involved a three-step decoction mashing process. Rice malt grist (~2 kg) was dissolved in clean filtered potable water (8 L) previously made to a pH of 11.0 using Ca(OH)2 solution. The entire mash temperature was raised to 35–40 °C to acidulate the mash followed by addition of 1 mL of protease for proteolysis to take place, with temperature maintained for 30 min at gentle stirring (acid rest). First decoction involved transferring one-third of the mash to a mash kettle and heated to 70 °C. Heated mash was transferred back to the remaining two-third mash, with entire temperature raised to 50–55 °C, followed by addition of α-amylase (0.8 mL) and subsequently allowed to rest for 30 min. In the second decoction, one-third of the mash was further heated in the mash kettle (3–5 min) until temperature of 85 °C was reached, then transferred to the remaining two-third thin mash, raising the temperature to ~67 °C and the mash gelatinized, which was allowed to rest for 30 min after addition of α-amylase (0.8 mL). Following liquefaction, a third decoction involved raising temperature to 100 °C, wherein boiled mash was added to the remaining mash, which moderated the entire temperature to about 72–75 °C. Mash rested again for ~30 min after addition of β-amylase (0.8 mL) prior to saccharification. To denature the enzymes prior to wort lautering, mashing-out was carried out. Spent grains were sparged with 1,000 mL of hot sparge water at 80 °C to obtain the entire wort from the mashing operation, before wort was concentrated.

Figure 2 Flow diagram for the production of rice malt wort.

Production of rice malt beer

In this study, the Japanese rice lager (beer style) production process was adopted (Briess, 2020; Beeradvocate, 2020) for the production of rice malt beer, following the method described by Briggs et al. (2004) with slight modification. Briefly, rice malt wort was boiled together with hop extracts for 30 min and thereafter, allowed to cool. Undissolved particles were removed and the filtered wort was transferred into fermenters. Already activated yeast (3 g) culture (Saccharomyces pastorianus) was pitched into the fermenting vessel at 20 °C and fermentation was carried out at 10–20 °C for 8 days. Green beer was filtered and allowed to age for 21 days. Matured/draft beer was siphoned into sterile bottles, pasteurized at 60 °C for 15 min, and subsequently, analysed.

Rice malt analyses

Malt analyses are carried out in accordance with standard methods of the Institute of Brewing (IoB), European Brewing Convention (EBC), and American Society of Brewing Chemists (ASBC) for several purposes such as to provide data for the maltster to use for quality control and to guide process adjustments, to provide a basis for product valuation, for prediction of extract recovery, to indicate the potential value of the malt and whether or not a particular malt is likely to give production difficulties (Briggs, 1998).

Determination of cold water extract

Cold water extract (CWE) measures the pre-formed water-soluble substances present in rice malt (Briggs, 1998). CWE (presented in g/100 g) was determined using method recommended by Institute of Brewing (IOB) (2007) and was calculated using equation below: (5) ColdWaterExtract(%)=G×1003.86×20

where,

G = dimensionless quantity depicting the excess degrees of gravity of the filtrate at 15.5 °C as 1,000

that is, G = 1,000 (SG – 1)

SG = Specific gravity

Determination of hot water extract

Hot water extract (HWE) measures the extractable materials derived from rice malt after a small scale mashing process (Briggs, 1998). HWE expressed as liter degrees/kg (L°/kg) was determined using method recommended by Institute of Brewing (IOB) (2007) and was calculated using the equation below; (6) HotWaterExtract(L°/kg)=G×VM

where,

G = dimensionless quantity depicting the excess degrees of gravity of the filtrate at 15.5 °C as 1,000

that is, G = 1,000 (SG – 1)

V = Volume of wort in Litres (L)

M = Mass of rice malt in Kilogram (kg)

SG = Specific gravity

Determination of diastatic power

Diastatic power (DP) measures the amount of enzyme in rice malt available to convert complex carbohydrates/starches into fermentable sugars (Ackley, 2018). DP expressed in lintner degree (°Lintner) was determined using method recommended by Institute of Brewing (IOB) (2007) and was calculated using the equation below; (7) DiastaticPower(°Lintner)=2000−200xy−xs

where:

x = Volume of rice malt extract (mL)

y = Volume of converted starch to 5 mL of the Fehling’s solution (mL)

s = Titre for starch blank (mL)

Determination of moisture content and total nitrogen

Rice malt with higher DP, CWE, and HWE were selected for wort production/brewing trial. Only these were subject to determinations of moisture content (MC) presented in g/100 g (wet basis) via the method described in AOAC (2006), and total nitrogen (TN) presented in g/L via the Kjeldahl method (European Brewery Convention, 2006).

Determination of malting loss

Malting loss (ML) after germination was determined according to the method described by Adebowal et al. (2010) by weighing the rice grains before and after malting. The weight of 100 grains of rice was recorded before malting and the weight of the malted grains after the rootlets were removed by hand was also recorded. ML was expressed as percentage (%) on dry matter basis.

(8) Maltingloss(%)=Weightofunmaltedgrain−WeightofmaltedgrainWeightofunmaltedgrain×100

Determination of malt yield

Malt yield (MY) after germination was determined according to the method described by Adebowal et al. (2010) by weighing the rice grains before and after malting. The weight of 100 grains of rice was recorded before malting and the weight of the malted grains after the rootlets were removed by hand was also recorded. Malt yields were expressed as percentage (%) on dry matter basis.

(9) Maltyield(%)=WeightofmaltedgrainWeightofunmaltedgrain×100

Rice wort analyses

Determination of pH

pH of rice malt wort was determined using the method described by AOAC (2004). Digital pH meter calibration used buffer 4, 7 and 9 solutions at 25 °C. pH conducted measurement required electrode (probe) dipping into each 25 mL pipetted wort sample, allowed to stabilize before reading off.

Determination of total soluble nitrogen

Total soluble nitrogen (TSN) measures the nitrogen materials (amino acids, peptides, and polypeptides) solubilized by proteolysis during malting and extracted during mashing (Agu & Palmer, 1998; Noonan, 2003). TSN of rice malt wort, expressed in g/L was determined using Kjeldhal method (Institute of Brewing (IOB), 2007).

Determination of apparent brix

Brix (°Bx) of rice malt wort was determined using a Milwaukee Digital brix Refractometer Model MA871 (Milwaukee Instruments, Rocky Mount, NC, USA) (Montañez-Soto et al., 2013), which involved refractometer standardized with distilled water at 20 °C until brix value read zero, followed by two drops of wort sample on the lens (sensitive surface), and measurement conducted subsequently.

Determination of kolbach index

Kolbach index (KI) measures the degree/extent of protein modification/degradation, as a ratio of TSN in wort to TN in the rice malt (Bamforth, 2003; Oliver & Colicchio, 2012). KI expressed in %, was calculated consistent with Analytical—EBC European Brewery Convention (1998) method, using the equation below: (10) KolbachIndex(%)=TSNTN×100

where TSN = Total soluble nitrogen (g/L)

TN = Total nitrogen (g/L)

Determination of free amino nitrogen

Free amino nitrogen (FAN) of rice malt wort was determined by Ninhydrin method (European Brewery Convention, 1998) with slight modifications. The sample (one mL) diluted with deionised water to 100 mL, then two mL of diluted sample mixed with one mL of colour reagent, placed in boiled water for 16 min, and allowed to cool to 20 °C. Diluted solution (five mL) was added, followed by measurement of optical density at 570 nm. Blank was determined with two mL of deionized water. Glycine standard solution was checked using two mL of glycine solution. The FAN content was calculated and expressed in mg/L.

Determination of dextrose equivalence

Dextrose equivalent (DE) measures the amount of reducing sugars present in a sugar product, relative to glucose (Dziedzic & Kearsley, 1995) determined on rice malt wort, using the Lane and Eynon Fehling’s solution method as previously described (International Starch Institute, 1999).

Determination of original extract

Original extract (OE) measures wort density compared to that of water at equal volume and temperature (ASBC, 2009). OE of rice malt wort presented in g/100 g was calculated from an approximate Plato value as previously described (Kunze, 2004), calculated using equation below: (11) SpecificGravity(SG)=(Masspercent×0.004)+1

where Mass percent ≡Apparent brix value (12) OriginalExtract(g/100g)=259–(259SG)

where SG = Specific gravity

Determination of wort sugar profile

The sugar profile of rice malt wort was determined using HPLC according to the method described in AOAC Official Method 982.14 AOAC (2006).

Rice malt beer analyses

Determination of pH, colour, apparent extract, alcohol and turbidity

The pH, colour, apparent extract (AE), alcohol and turbidity was evaluated using the following Analytical-EBC methods (European Brewery Convention, 2007). pH was determined using the EBC method 9.35, similar to AOAC (2004). Colour presented in °EBC was determined using the Spectrophotometric Method (EBC method 9.6). AE presented in g/100 g was determined using the EBC method 9.43.1. Alcohol by volume (%ABV) was determined by distillation (EBC method 9.2.1). Turbidity presented in Nephelometric Turbidity Unit (NTU) was determined by EBC method 9.10.

Determination of sensory attributes

A hedonic scale test was used to evaluate the sensory attributes of beer samples according to the method reported by Iwe (2002). This specifically involved comparing the rice malt beer samples with the commercial lager beer. The sensory evaluation was carried out by 20 ordinary frequent beer drinkers (semi-trained panelists) of different age groups (20–36 years old). The prerequisites for participating in the study were that the individual consumed beer and showed interest in participating in all test sessions. Importantly, the participation in this sensory evaluation was voluntary, and oral consent was obtained prior to participation. The coded rice malt beer samples were randomly served at temperatures of about 10 °C. Participants were served with four series of beer samples in transparent glass cups and the degree of liking was rated using a nine-point hedonic scale with the ratings of 9 as liked extremely and 1 as disliked extremely for five main attributes that is, colour, aroma, taste, mouthfeel, and appearance; while overall acceptance of the samples was evaluated by taking the average of other attributes. The panelists drank potable water to rinse/clean their mouth between tastings to avoid cross-contamination between samples. After tasting, score sheets were filled by the tasters.

Statistical analysis

One-way analysis of variance (ANOVA) was carried out using the IBM SPSS version 20 Software (IBM, New York, NY, USA). Results were expressed as mean standard deviation (SD). Mean differences were resolved using the least significant differences (LSD) at post-hoc conditions. The level of statistical significance was considered at p < 0.05.

Results

Changes in grain characteristics of rice varieties

The grain characteristics of rice varieties is shown in Table 1. Specifically, the grain characteristics were depicted in terms of TKW (g), GE (%), GC (%), and DoS (%). Results showed that TKW, GE, GC, and DoS varied significantly (p <0.05) across the rice varieties. Further, the TKW ranged between 25.81 g and 27.01 g, GE ranged between 86.50% and 95.00%, GC ranged between 92.50% and 95.50%, and DoS ranged between 49.96% and 55.00%. Clearly, the TKW peaked at NERICA 7 (27.01 g). Additionally, both GE and GC peaked at NERICA 7, with 95.00% and 95.50% respectively. Moreover, the DoS was peak (55.00%) at FARO 44, and least at NERICA 7 (49.96%). Besides, the rice varieties obtained acceptable grain germination of above 85%.

Table 1 Grain characteristics of rice varieties.

SAMPLES	TKW (g)	GE (%)	GC (%)	DoS (%)	
FARO 44	25.81c ± 0.01	86.50c ± 0.06	92.50b ± 0.14	55.00a ± 0.16	
FARO 57	26.11b ± 0.16	93.50b ± 0.07	93.00b ± 0.00	50.98b ± 0.07	
NERICA 7	27.01a ± 0.07	95.00a ± 0.98	95.50a ± 0.10	49.96c ± 0.09	
LSD	0.24	1.50	1.50	0.44	
Notes:

a–cThe same superscript letter along a column for each parameter is not significantly different (p > 0.05).

Values are the means of duplicate determinations (N = 2).

TKW, thousand kernel weight; GE, germination energy; GC, germinative capacity; DoS, degree of steeping.

Changes in CWE, HWE, DP, MY, ML, MC and TN of rice malt

The CWE (g/100 g), HWE (L°/kg), and DP (°Lintner) of rice malts subject to varying steeping durations, germination periods, and kilning temperatures are shown in Tables 2–4, respectively. Results showed that CWE, HWE, and DP in rice malt varied significantly (p < 0.05) across the varieties. Specifically, the CWE ranged between 10.95 and 23.16 g/100 g, HWE ranged between 51.06 and 206.48 L°/kg, and DP ranged between 22 and 150 °Lintner, across rice varieties. With respect to S18G3K55° (18 h steeping, 3 days germination, 55 °C kilning), S30G2K50° (30 h steeping, 2 days germination, 50 °C kilning), and S24G3K55° (24 h steeping, 3 days germination, 55 °C kilning) combinations, the CWE, HWE, and DP obtained peaks at FARO 44 (22.88 g/100 g; 136.56 L°/kg; 150 °Lintner), FARO 57 (22.13 g/100 g;170.18 L°/kg; 148 °Lintner), and NERICA 7 (23.16 g/100 g; 206.48 L°/kg; 143 °Lintner). Based on these CWE/HWE/DP peaks, the SGK (steeping duration, germination periods and kilning temperatures) combinations of rice varieties trended as follows: CWE = NERICA 7 (23.16 g/100 g) > FARO 44 (22.88 g/100 g)> FARO 57 (22.13 g/100 g); HWE = NERICA 7 (206.48 L°/kg) > FARO 57 (170.18 L°/k) > FARO 44. (136.56 L°/kg); and DP = FARO 44 (150 °Lintner) > FARO 57 (148 °Lintner) > NERICA 7 (143 °Lintner). Outside this specific SGK combinations, the CWE/HWE/DP obtained no peaks.

Table 2 The CWE (g/100 g) of rice malts subject to varying steeping durations, germination periods and kilning temperatures.

Malting conditions	Rice varieties	
Steeping duration (hours)	Germination period (days)	Kilning temperature (°C)	FARO 44	FARO 57	NERICA 7	
18	2	50	21.66d ± 0.03	19.43c ± 0.28	10.95l ± 0.01	
55	22.28b ± 0.06	14.39j ± 0.42	12.68i ± 0.04	
3	50	21.84c ± 0.10	13.99k ± 0.06	11.73k ± 0.06	
55	22.88a ± 0.13	11.44m ± 0.06	13.31f ± 0.07	
4	50	20.45j ± 0.08	16.51h ± 0.07	12.68i ± 0.06	
55	20.53hi ± 0.17	14.38j ± 0.11	20.25c ± 0.17	
24	2	50	20.77f ± 0.14	20.58b ± 0.14	12.52j ± 0.08	
55	20.95e ± 0.14	15.17i ± 0.10	14.94d ± 0.10	
3	50	20.58gh ± 0.11	18.24e ± 0.10	12.52j ± 0.07	
55	20.76f ± 0.06	13.20l ± 0.04	23.16a ± 0.07	
4	50	19.81l ± 0.08	16.91g ± 0.07	12.93h ± 0.57	
55	20.49ij ± 0.08	17.18f ± 0.07	21.13b ± 0.08	
30	2	50	18.50n ± 0.04	22.13a ± 0.06	13.32f ± 0.03	
55	19.72m ± 0.13	16.51h ± 0.11	17.19g ± 0.11	
3	50	19.61k ± 0.14	18.98d ± 0.13	13.50f ± 0.10	
55	20.58g ± 0.03	15.17i ± 0.03	14.54d ± 0.01	
4	50	18.49n ± 0.06	19.03d ± 0.06	13.69f ± 0.04	
55	20.46ij ± 0.08	16.51h ± 0.07	16.34f ± 0.07	
LSD			0.27	0.59	0.45	
Notes:

a–nThe same superscript letter along a column for each rice variety is not significantly different (p > 0.05).

Values are the means of duplicate determinations (N = 2).

Table 3 The HWE (L°/kg) of rice malts subject to varying steeping durations, germination periods, and kilning temperatures.

Malting conditions	Rice varieties	
Steeping duration (h)	Germination period (days)	Kilning temperature (°C)	FARO 44	FARO 57	NERICA 7	
18	2	50	121.56b ± 0.48	103.05h ± 0.41	130.06h ± 0.54	
55	103.18e ± 1.02	103.18gh ± 0.99	182.10b ± 1.80	
3	50	103.18e ± 0.74	105.17f ± 0.75	163.71e ± 1.16	
55	136.56a ± 1.16	100.15i ± 0.88	182.10b ± 1.54	
4	50	106.22d ± 0.59	160.43c ± 0.91	121.41k ± 0.69	
55	115.56c ± 0.81	105.17f ± 0.74	163.90e ± 1.13	
24	2	50	100.15f ± 1.13	106.22e ± 1.20	166.91d ± 1.90	
55	84.97j ± 0.83	106.22e ± 1.02	148.00f ± 1.47	
3	50	100.15f ± 0.17	106.22e ± 0.18	182.10b ± 0.31	
55	91.04i ± 0.44	103.18gh ± 0.51	206.48a ± 0.99	
4	50	84.97j ± 0.54	106.22e ± 0.68	182.10b ± 1.16	
55	106.18d ± 0.28	103.48g ± 0.28	133.70i ± 0.35	
30	2	50	51.06m ± 0.33	170.18a ± 1.10	139.60g ± 0.91	
55	75.98k ± 0.24	136.56d ± 0.34	127.50j ± 0.41	
3	50	75.98k ± 0.65	166.91b ± 1.41	127.50j ± 1.08	
55	97.06g ± 0.55	106.22e ± 0.78	163.90e ± 0.93	
4	50	60.70l ± 0.21	106.02e ± 0.35	172.98c ± 0.59	
55	94.21h ± 0.62	103.08h ± 0.68	133.70i ± 0.89	
LSD			0.49	0.30	1.19	
Notes:

a–lThe same superscript letter along a column for each rice variety is not significantly different (p > 0.05).

Values are the means of duplicate determinations (N = 2).

Table 4 The DP (°Lintner) of rice malts subject to varying steeping durations, germination periods and kilning temperatures.

Malting conditions	Rice varieties	
Steeping duration (h)	Germination period (days)	Kilning temperature (°C)	FARO 44	FARO 57	NERICA 7	
18	2	50	78.00e ± 0.74	63.00f ± 0.59	71.00j ± 0.66	
55	92.00c ± 0.59	22.00n ± 0.14	97.00g ± 0.61	
3	50	63.00g ± 0.28	50.00i ± 0.23	48.00m ± 0.21	
55	150.00a ± 0.23	38.00l ± 0.06	60.00l ± 0.10	
4	50	80.00d ± 0.37	37.00l ± 0.17	67.00k ± 0.30	
55	98.00b ± 0.42	26.00m ± 0.11	100.00e ± 0.44	
24	2	50	43.00l ± 0.06	80.00e ± 0.11	98.00f ± 0.14	
55	63.00f ± 0.18	39.00k ± 0.11	120.00c ± 0.34	
3	50	50.00j ± 0.21	64.00f ± 0.27	60.00l ± 0.25	
55	52.00i ± 0.28	52.00h ± 0.28	143.00a ± 0.79	
4	50	44.00k ± 0.34	57.00g ± 0.44	60.00l ± 0.45	
55	57.00h ± 0.61	39.01k ± 0.41	101.00e ± 1.08	
30	2	50	38.00p ± 0.15	148.00a ± 0.58	102.00e ± 0.41	
55	39.00o ± 0.38	43.00j ± 0.42	113.00d ± 1.12	
3	50	35.00q ± 0.25	135.00b ± 0.96	92.00h ± 0.65	
55	38.00p ± 0.32	98.08c ± 0.83	138.00b ± 1.17	
4	50	39.00o ± 0.23	92.00d ± 0.52	46.00m ± 0.25	
55	40.00m ± 0.24	43.00j ± 0.25	75.00i ± 0.45	
LSD			0.73	0.98	1.63	
Notes:

a–qThe same superscript letter along a column for each rice variety is not significantly different (p > 0.05).

Values are the means of duplicate determinations (N = 2).

The MY, ML, MC, and TN of rice malt samples is shown in Table 5. Results showed that ML and MY of rice malt changed significantly (p < 0.05) across the varieties. The MY ranged between 87.11% and 92.65%, whereas the ML ranged between 6.03% and 10.80%. Additionally, the peak (10.80%) ML and least (87.11%) MY can be seen at NERICA 7, whereas the peak MY (92.65%) and least ML (6.03%) can be seen at FARO 44. Specifically from FARO 44 (S18G3K55°), FARO 57 (S30G2K50°), and NERICA 7 (S24G3K55°), the MC and TN of rice malts were determined (also shown in Table 5). Results showed that both MC and TN varied significantly (p < 0.05). Specifically, the MC of rice malts ranged between 5.19 and 6.43 g/100 g, whereas the TN of rice malts ranged between 13.10 and 15.70 g/L. Additionally, the MC in rice varieties trended as follows: NERICA 7 (6.43 g/100 g) > FARO44 (5.52 g/100 g) > FARO57 (5.19 g/100 g), whereas the TN in rice varieties trended as follows: FARO 44 (15.70 g/L) > FARO 57 (14.30 g/L) > NERICA 7 (13.10 g/L). Additionally, the peaks of MC and TN were obtained at NERICA 7 (6.43 ± 0.14 g/100 g) and FARO 44 (15.70 ± 0.05 g/L), respectively.

Table 5 Malt yield (MY), malting loss (ML), moisture content (MC) and total nitrogen (TN) of rice malt samples.

Samples	MY (%)	ML (%)	MC (g/100 g)	TN (g/L)	
FARO 44	92.65a ± 0.33	6.03c ± 0.62	5.52b ± 0.18	15.70a ± 0.05	
FARO 57	90.65b ± 0.95	8.05b ± 0.38	5.19c ± 0.13	14.30b ± 0.17	
NERICA 7	87.11c ± 0.74	10.80a ± 0.25	6.43a ± 0.14	13.10c ± 0.16	
LSD	0.34	0.05	0.28	0.24	
Notes:

a–cThe same superscript letter along a column for each rice variety is not significantly different (p > 0.05).

Values are the means of duplicate determinations (N = 2).

Changes in sugar profile, pH, TSN, Brix, KI, FAN, DE, and OE of rice malt wort

The sugar profile of rice malt wort samples is shown in Table 6. A combination of sugars (glucose, maltose, maltotetraose, maltotriose, raffinose, and sucrose) can be seen, which significantly differed (p < 0.05) across varieties. In particular, the glucose ranged between 10.84% and 11.63%, maltose ranged between 14.63% and 15.34%, maltotetraose ranged between 0.44% and 0.63%, maltotriose ranged between 12.26% and 16.40%, raffinose ranged between 0.05% and 0.07%, and sucrose ranged between 2.32% and 2.83%. Additionally, the sugars in rice malt wort trended by varieties as follows: glucose = FARO 57 > FARO 44 > NERICA 7; maltose = FARO 57 > NERICA 7 > FARO 44; maltotetraose = FARO 44 > FARO 57 > NERICA 7; maltotriose = NERICA 7 > FARO 57 > FARO 44; raffinose = FARO 44 > NERICA 7 > FARO 57; and sucrose = NERICA 7 > FARO 44 > FARO 57. From these, we see that FARO 57 obtained peaks at glucose (11.63 ± 0.71%) and maltose (15.34 ± 0.08%), FARO 44 obtained peaks at maltotetraose (0.63 ± 0.04%) and raffinose (0.07 ± 0.00%), NERICA 7 obtained peaks at maltotriose (16.40 ± 0.07%) and sucrose (2.83 ± 0.08%).

Table 6 Sugar profile of rice malt wort samples.

SAMPLES	Maltotriose (%)	Glucose (%)	Maltose (%)	Maltotetraose (%)	Sucrose (%)	Raffinose (%)	
FARO 44	12.26c ± 0.28	11.23a ± 0.16	14.63b ± 0.11	0.63a ± 0.04	2.44b ± 0.10	0.07a ± 0.00	
FARO57	13.25b ± 0.14	11.63a ± 0.71	15.34a ± 0.08	0.53b ± 0.03	2.32c ± 0.03	0.05b ± 0.00	
NERICA 7	16.40a ± 0.07	10.84b ± 0.06	15.03a ± 0.04	0.44c ± 0.06	2.83a ± 0.08	0.06ab ± 0.00	
LSD	0.03	0.02	0.03	0.03	0.03	0.01	
Notes:

a–cThe same superscript letter along a column for each rice variety is not significantly different (p > 0.05).

Values are the means of duplicate determinations (N = 2).

The pH, TSN (g/L), Brix (g/100 g), KI (%), FAN (mg/L), DE (g/100 g), and OE (g/100 g) components of rice malt wort samples is shown in Table 7. Results showed that, whereas Brix, KI, and FAN varied significantly (p < 0.05), the pH, TSN, DE, and OE resembled (p > 0.05), with the following ranges across rice varieties: pH ranged between 5.30 and 5.40, TSN ranged between 5.40 and 5.80 g/L, Brix ranged between 13.88 and 16.36 g/100 g, KI ranged between 34.39% and 44.27%, FAN ranged between 108.56 and 117.34 mg/L, DE ranged between 37.00 and 40.00 g/100 g, and OE ranged between 9.68 and 12.66 g/100 g. By rice varieties, therefore, these (above-mentioned) parameters in rice malt wort obtained peaks at FARO 44 (pH = 5.40), NERICA 7 (TSN = 5.80 g/L), FARO 57 (Brix = 16.36 g/100 g), NERICA 7(KI = 44.27%), NERICA 7 (FAN = 117.34 mg/L), and FARO 57 (DE = 40.00 g/100 g)/ (OE = 12.66 g/100 g). Additionally, these parameters of rice malt wort trended as follows: pH = FARO 44 (5.40) > FARO 57/NERICA 7 (5.30); TSN = NERICA 7 (5.80 g/L) > FARO 57 (5.60 g/L) > FARO 44 (5.40 g/L); Brix = FARO 57 (16.36 g/100 g) > FARO 44 (14.65 g/100 g) > NERICA 7 (13.88 g/100 g); KI = NERICA 7 (44.27%) > FARO 57 (39.16%) > FARO 44 (34.39%); FAN = NERICA 7 (117.34 mg/L) > FARO 57 (112.23 mg/L) > FARO 44 (108.56 mg/L); DE = FARO 57 (40 g/100 g) > FARO 44 (39 g/100 g) > NERICA 7 (37 g/100 g); and OE = FARO 57 (12.66 g/100 g) > FARO 44 (11.15 g/100 g) > NERICA 7 (9.68 g/100 g).

Table 7 The pH, TSN (g/L), Brix (g/100 g), KI (%), FAN (mg/L), DE (g/100 g) and OE (g/100 g) components of rice malt wort samples.

Samples	pH	TSN (g/L)	Brix (g/100 g)	KI (%)	FAN (mg/L)	DE (g/100 g)	OE (g/100 g)	
FARO 44	5.40a ± 0.38	5.40a ± 0.00	14.65b ± 0.07	34.39c ± 0.13	108.56c ± 0.08	39.00a ± 1.42	11.15a ± 0.05	
FARO 57	5.30a ± 0.17	5.60a ± 0.00	16.36a ± 0.42	39.16b ± 0.09	112.23b ± 0.28	40.00a ± 0.82	12.66a ± 0.04	
NERICA 7	5.30a ± 0.10	5.80a ± 0.00	13.88c ± 0.11	44.27a ± 0.28	117. 34a ± 0.06	37.00a ± 0.01	9.68a ± 0.07	
LSD	NS	NS	0.58	3.20	4.32	NS	NS	
Notes:

a–cThe same superscript letter along a column for each rice variety is not significantly different (p > 0.05).

Values are the means of duplicate determinations (N = 2).

TSN, total soluble nitrogen; KI, kolbach index; FAN, free amino nitrogen; DE, dextrose equivalent and OE, original extract

Changes in pH, colour, AE, alcohol content, turbidity and sensory attributes of rice malt beer

The pH, colour (°EBC), AE (g/100 g), alcohol content (%ABV), and turbidity (NTU) of rice malt beer samples is shown in Table 8. Results showed that, whereas the alcohol content varied significantly (p < 0.05), the pH, colour, AE, and turbidity resembled (p > 0.05), with the following ranges across rice varieties: pH ranged between 3.80 and 3.90, colour ranged between 3.20 and 3.73 °EBC, AE ranged between 4.57 and 4.93 g/100 g, alcohol content ranged between 2.82 and 4.13 %ABV and turbidity ranged between 4.30 and 4.80 NTU. By rice varieties, therefore, these (above-mentioned) parameters in rice malt beer obtained peaks at FARO 57 (pH = 3.90; colour = 3.73 °EBC; and alcohol content = 4.13 %ABV), NERICA 7 (AE = 4.93 g/100 g) and FARO 44 (turbidity = 5.30 NTU). Additionally, these parameters of rice malt beer trended as follows: pH = FARO 57 (3.90) > FARO 44 and NERICA 7 (3.80); colour = FARO 57 (3.73 °EBC) > FARO 44 (3.70 °EBC) > NERICA 7 (3.20 °EBC); AE = NERICA 7 (4.93 g/100 g) > FARO 44 (4.59 g/100 g) > FARO 57 (4.57 g/100 g) ; alcohol content = FARO 57 (4.13%ABV) > FARO 44 (3.54%ABV) > NERICA 7 (2.82%ABV) and turbidity = FARO 44= (5.30 NTU)> FARO 57 (4.80 NTU) > NERICA 7 (4.30 NTU).

Table 8 The pH, colour (°EBC), AE (g/100 g), alcohol content (%ABV) and turbidity (NTU) of rice malt beer samples.

Samples	pH	Colour (°EBC)	AE
(g/100 g)	Alcohol content
(% ABV)	Turbidity
(NTU)	
FARO 44	3.80a ± 0.44	3.70a ± 0.16	4.59a ± 0.74	3.54b ± 0.06	5.30a ± 0.01	
FARO 57	3.90a ± 0.59	3.73a ± 0.71	4.57a ± 1.07	4.13a ± 0.18	4.80a ± 0.17	
NERICA 7	3.80a ± 0.10	3.20a ± 0.42	4.93a ± 0.54	2.82c ± 0.50	4.30a ± 0.21	
LSD	NS	NS	NS	0.39	NS	
Notes:

a–cThe same superscript letter along a column for each rice variety is not significantly different (p > 0.05).

Values are the means of duplicate determinations (N = 2).

AE, apparent extract; ABV, alcohol by volume

The sensory attributes (colour, aroma, taste, mouthfeel, appearance, and overall acceptability) of rice malt beer samples is shown in Table 9. Importantly, the sensory attributes of rice malt beer was compared with the commercial lager beer. Results showed that colour, taste, mouthfeel, appearance, and overall acceptability of rice malt beer were significantly (p < 0.05) less than those of commercial lager beer. Only the aroma of rice malt beer resembled (p > 0.05) that of commercial lager beer. Specific to the rice malt beer, the sensory attributes ranged as follows: the colour ranged between 6.66 and 6.91; aroma ranged between 7.54 and 7.81; mouthfeel ranged between 6.57 and 6.96; appearance ranged between 6.24 and 6.52; taste ranged between 7.69 and 7.87, and overall acceptability ranged between 6.94 and 7.21. By rice varieties with respect to this (rice) malt beer, only the sensory attributes of FARO 44 obtained peaks, namely: colour = 6.91; taste = 7.87; aroma = 7.81; mouthfeel = 6.96; appearance = 6.52 and overall acceptability =7.21.

Table 9 The sensory attributes (colour, aroma, taste, mouthfeel, appearance and overall acceptability) of rice malt beer samples.

SAMPLES	Colour	Taste	Aroma	Mouthfeel	Appearance	Overall acceptability	
Commercial lager beer	8.71a ± 0.55	8.19a ± 0.13	7.99a ± 0.51	8.51a ± 0.25	8.61a ± 0.45	8.40a ± 0.38	
FARO 44	6.91b ± 0.33	7.87b ± 0.32	7.81a ± 0.33	6.96b ± 0.32	6.52b ± 0.32	7.21b ± 0.30	
FARO 57	6.66b ± 0.13	7.69b ± 0.21	7.54a ± 0.16	6.57b ± 0.41	6.24b ± 0.23	6.94b ± 0.08	
NERICA 7	6.87b ± 0.25	7.82b ± 0.10	7.80a ± 0.51	6.76b ± 0.22	6.41b ± 0.23	7.13b ± 0.24	
LSD	0.85	NS	NS	1.22	0.93	0.56	
Notes:

a–bThe same superscript letter along a column for each rice variety is not significantly different (p > 0.05).

Values are the means of duplicate determinations (N = 2).

Discussion

Discussion of rice grain

Across the varieties, the TKW rice grain range (25.81–27.01 g) (Refer to Table 1) competes well with those reported by Osuji, Ofoedu & Ojukwu (2019), attributable to differences in soil composition, weather condition, moisture content, or grain production/harvest period. TKW could help identify grain/seed density, size, and variety (Tokpah, 2010; Osuji, Ofoedu & Ojukwu, 2019). Potentially, the NERICA 7 might have a higher starch content compared to other rice varieties at this current study. Higher TKW or large grain size as found in NERICA 7 (Refer to Table 1) could be indicative of high starch content (Ayernor & Ocloo, 2007). The GE (86.50–95.00%) and GC (92.50–95.50%) of rice grain (Refer to Table 1) suggests it a very promising substitute for barley in brewing, and probably very viable during malting (Adebowal et al., 2010; Osuji, Ofoedu & Ojukwu, 2019). Such variations in GE and GC across the studied rice varieties might be due to influences of (rice) harvest period, kernel size, starch content, and water absorption rates. Grain germination of above 90% portrays a good quality malt attribute (Agbale et al., 2007). Rice varieties obtained acceptable grain germination of above 85%, which makes it acceptable for malting purposes. Similar trends of over 85% germinative properties are reported elsewhere (Bam et al., 2006; Hammond & Ayernor, 2001; Ameko et al., 2013). Higher GE enhances the enzyme activities as well as seed vigour (Agbale et al., 2007). Besides, grain and malting characteristics can help ascertain the cereal as an acceptable substitute for barley.

The DoS, which is well-known as the amount of moisture/water absorbed by the grain particularly during the steeping process, remains an integral step in the malting process and accompanied by enzyme development and its associated metabolic influences. In the current work, the DoS was highest (55.00%) at FARO 44, and least at NERICA 7 (49.96%). Results showed that rice varieties with higher DoS obtained higher MY and lower ML (Refer to Table 1), probably due to decreased metabolic processes in the rice varieties of the current study. Besides small kernels taking up more water compared to larger kernels, it is also believed that the grains from inland regions would swell and germinate faster compared to grains from maritime regions (Kunze, 2005).

Discussion of rice malt

Rice malt’s CWE (10.95–23.16 g/100 g) range of current work (Refer to Table 2) agrees with those reported by Kasetsart (2007) and represents a ‘good modification’ based on CWE (15–22 g/100 g) range data of Briggs (1998) and Briggs et al. (2004). By hydrating the grist during grain modification, cold mashing (20 °C) solubilises enzymatically-degraded compounds (Dahiya et al., 2018). According to European Brewing Convention (EBC) and American Society of Brewing Chemists (ASBC), HWE range of 51.06–206.48 L°/kg (Refer to Table 3) will be equivalent to ~13–54 g/100 g soluble extract (SE). This would suggest the HWE of current work to be greater than two fold of CWE. Moreover, DP of rice malts (23–150 °Lintner) (Refer to Table 4) corroborate with barley (50–150 °Lintner) (BYO, 2019a), (35–40 °Lintner) (O’Rourke, 2002a), and sorghum (20–23 °L) malts (Byrne, Donnelly & Carrol, 1993). Malt DP range 35–40 °Lintner can convert its own starches (BYO, 2019b) probably with a longer conversion time. Malt enzymes that degrade starch and obtain high extract yield depict good malt characteristics (Subramanian et al., 1995; Muoria & Bechtel, 1998).

The CWE, HWE, and DP of rice malt increased with steeping duration at FARO 44 (S18G3K55°), FARO 57 (S30G2K50°), and NERICA 7(S24G3K55°) (Refer to Tables 1–3). DP’s significant role in HWE of rice malt, corroborates with data of millet malt (Eneje, Odibo & Nwani, 2012). Besides small-sized kernel of cereals modifying at a faster rate over large ones (Agu, 2009), the grain physiological activities progressing during malting (Kunze, 2004; Ogbonna, 2002; Osuji, Ofoedu & Ojukwu, 2019) could influence CWE, HWE, and DP of rice malt. Starch-degrading enzymes like alpha-amylase, beta-amylase, limit dextrinase and alpha-glucosidase (Buchholz, Volker & Uwe, 2005; Evans, Li & Eglinton, 2010), lipases, proteases and other enzymes (Briggs, 1998) could also influence DP of rice malt. Low DP of rice malts could corroborate with lower protein content of grain (Agu & Palmer, 1998). Mashing schedule, high gelatinization temperature, and kilning/malting conditions might be contributing to the differences in CWE, HWE, and DP of rice malt in this study.

To a maltster, MY is an important attribute because it gives an indication of the amount of recoverable soluble extracts from the malted grain (Osuji, Ofoedu & Ojukwu, 2019). The analysis of malt quality guides the maltster/brewer on variety selection and effectiveness of the malting process to optimise output, as well as to achieve sustainable malt brewing process (Briggs, 1998). On the other hand, ML measures the metabolic activity associated with grain germination, which increases with the germination period. In the current work, the MY (87.11–92.65%) and ML (6.03–10.80%) of rice malt changed significantly (p < 0.05) across varieties (Refer to Table 5) with peaks seen at NERICA 7 (ML = 10.80%) and FARO 44 (MY = 92.65%). Probably, these variations in ML and MY might have been influenced by the malting process (Ofoedu, 2018; Osuji, Ofoedu & Ojukwu, 2019). Besides moisture loss during kilning as well as physiological activities associated with germination, the changes in TKW, GE, and GC could also be influenced by both MY and ML. We hold this opinion, given the peak TKW, GE, and GC values obtained at NERICA 7, as well as the least MY obtained at FARO 44 (Comparing Tables 1 and 5).

The MC of rice malt (5.19–6.43 g/100 g) (Refer to Table 5) was above those of Munich (3.0–4.8 g/100 g) as well as two-row (2.0–4.3 g/100 g) barleys (Noonan, 2003). An increase in MC of malts can decrease the extract potential, which might lower the original gravity of the wort (BYO, 2019a). Malt closer to 1.5 g/100 g MC would be of less risk to mould growth (Noonan, 2003). A decrease in the moisture level of grain can be achieved with increased drying temperature(s) as well as prolonged drying time(s) (Lewis & Young, 2002; Osuji, Ofoedu & Ojukwu, 2019). Whilst low MC can prolong food shelf life (Alozie et al., 2009), a high MC can enhance its microbial spoilage (Ijarotimi, 2012). TN of rice malt (13.10–15.70 g/L) (Refer to Table 5) fell within those of ale/lager (14.00–18.00 g/L) (O’Rourke, 2002a), and sorghum (14.70–17.40 g/L) types (Agu & Palmer, 1998). Rice variety as well as malting conditions might also be contributing to the MC and TN differences in rice malt of this current work. Besides the amino acids being required for yeast growth, the hydrophobic nitrogen (from malt) provides foam and head retention in beer (O’Rourke, 2002a).

Discussion of rice malt wort

The importance of sugar wort composition/parameter to the brewer especially for fermentation cannot be over-emphasized. The sugar profile of rice malt wort is the outcome of enzymatic activities during mashing. In the current work, the rice malt wort yielded a combination of sugars, such as maltose (14.63–15.34%), maltotriose (12.26–16.40%), glucose (10.84–11.63%), sucrose (2.32–2.83%), raffinose (0.05–0.07%) and maltotetraose (0.44–0.63%), all of which varied significantly (p < 0.05) across varieties (Refer to Table 6). Clearly, it was not difficult to differentiate the sugars herein based on the amounts obtained across the studied rice varieties. Specifically, whereas the maltose, glucose and maltotriose clearly obtained higher amounts, the sucrose, maltotetraose, and raffinose obtained lower amounts. Both maltose and maltotriose, well-known as the predominant sugars found in wort seemed to be noticeably less than the values obtained by Ofoedu, Osuji & Ojukwu (2019).

There is a high chance that the conditions, mashing program as well as nature/type of (exogenous) enzymes used in this current work could have some impact on the variations in the rice malt wort sugar profile. Additionally, the lower sugar concentrations in the rice malt wort might be attributable to the limit dextrins likely produced in higher amounts, and maybe, tannins binding with malt’s amylase enzyme (Okolo et al., 2010). Sucrose, which is among the major soluble sugars and natural components of the matured kernel, was neither produced during malting nor hydrolysis/mashing, but however, could be depleted naturally during germination in sustaining (rice) malt metabolism. This might explain the significantly low sucrose concentration in the rice malt wort. The presence of maltotetraose and raffinose in the wort, however, could be indicative of oligosaccharides resulting from limited dextrins formation due to the different amylolytic enzymes working in the rice malts (Marconi et al., 2017).

Resembling (p > 0.05) across rice varieties (FARO 44 = 5.40 > FARO57/NERICA 7 = 5.30), the pH of rice malt wort (Refer to Table 7) compares well with a previously published (rice wort) data (4.98–6.08) (Sanchez-Moreno et al., 2006; Kasetsart, 2007) but slightly below those of barley ale/lager (5.6–5.9) (Palmer, 2006). During mashing and wort boiling, heat treatment can dissociate the calcium ion (Ca2+) bound with both phosphates (K2PO4) and polypeptides, forming insoluble compounds, releasing hydrogen ion (H+), and decreasing wort pH (Palmer, 2006). Increased wort acidity enhances both protein coagulation and yeast growth, and inhibits microbial contamination (O’Rourke, 2002b).

The TSN of rice malt wort which resembled (p > 0.05) across rice varieties (NERICA 7 = 5.80 g/L > FARO 57 = 5.60 g/L > FARO 44 = 5.40 g/L) (Refer to Table 7) corroborates favourably with those of sorghum (5.00–7.00 g/L) (Agu & Palmer, 1998) and typical lager barleys (5.70–6.60 g/L) (O’Rourke, 2002a). Aided by denaturation and precipitation of solubilized proteins, high gelatinization temperature of rice starch reduces TSN level in the wort (Jones, 1999). Steeping could enhance the loss of some soluble nitrogenous compounds, like amino acids (Briggs, 1998). Amino acid dissolution could increase TSN (during germination) owed to increased activity of protease enzyme (Banusha & Vasantharuba, 2013), which would cease if acrospires reach from 3/4 to 7/8 of grain length (Briggs et al., 2004). As the need for wort TSN increases, it becomes undesirable when protein degradation raises the TSN levels higher than required, thereby causing a reduction in foam formation, abnormal fermentation (Sadosky, 2007), and haze formation (Briggs et al., 2004).

The Brix of rice malt wort differed significantly (p < 0.05) across rice varieties (FARO 57 = 16.36 g/100 g > FARO 44 = 14.65 g/100 g > NERICA 7 = 13.88 g/100 g) (Refer to Table 7). A peak Brix at FARO 57 suggested increased malting accessibility to the substrate (starch) with enhanced enzymatic hydrolysis. Grain kernel size differences in rice varieties might affect the endosperm starch composition when malted/mashed, which may well vary the Brix values. Varying malting conditions influence the degree of grain response to modification, which could actually differ amongst rice varieties during germination. Brix value could, therefore, be affected by the variety and amount/type of sugars in the wort, which serves as a nutrient for yeast (Pedley, 1996).

The KI of rice malt wort differed significantly (p < 0.05) across rice varieties (NERICA 7 = 44.27% > FARO 57 = 39.16% > FARO 44 = 34.39%) (Refer to Table 7) with its range (34.39–44.27%) resembling those of typical lager malt (34–44%) (O’Rourke, 2002a). Malts can be classified based on the degree of modification (BYO, 2019b), namely: (a) under modified (KI values between 30% and 35%), (b) well modified (KI values above 35%), (c) over modified (KI values above 45%) malts. Specifically, wort KI of FARO 44 (34.39%) falls within ‘under modified’, whereas FARO 57 (39.16%) and NERICA 7 (44.27%) falls within ‘well modified’ malts. Further, Bamforth (2003) reported ‘well modified’ malt with KI range of 38–42%. Besides malting enabling KI to increase with germination, the small-size of FARO 44 kernel may corroborate with lower (KI) value. Thinner kernel/grain size taking up water faster (Kunze, 2004; Osuji, Ofoedu & Ojukwu, 2019) might sustain a higher TN relative to the larger ones (Briggs, 1998; Broadbent & Palmer, 2001; Koliatsou & Palmer, 2003). The reduced HWE, CWE, and KI values might help in defining those of FARO 44 as ‘under modified malt’. Howbeit, NERICA 7 higher TSN (5.80 g/L) and KI values (44.27%) would suggest a positive association of grain size with HWE, CWE, and TSN of the current study.

The FAN of rice malt wort differed significantly (p < 0.05) across rice varieties (NERICA 7 = 117.34 mg/L > FARO 57 = 112.23 mg/L > FARO 44 = 108.56 mg/L) (Refer to Table 7), and compared well with those of sorghum (94–216 mg/L) (Agu & Palmer, 1998), maize (100–169 mg/L), and rice (95–138 mg/L) (Taylor, Dlamini & Kruger, 2013) malts. Increased amino acids/protein modification might favour the peak FAN in NERICA 7 with proteolytic enzyme activity. As the principal nitrogen source in the wort, FAN depicts hydrolysed (soluble) proteins during mashing (Agu & Palmer, 1998; Russell, 2006), summed up by amino acids, ammonium ions, and small peptides (dipeptides and tripeptides) (Stewart, Hill & Lekkas, 2013; Lekkas et al., 2005; Pugh, Maurer & Pringle, 1997). Typical lager malt with FAN between 100 and 140 mg/L can enhance both efficient yeast cell growth and fermentation performance (Lekkas et al., 2005), in order to achieve a trouble-free fermentation (Briggs et al., 2004). Besides, FAN can also help to predict yeast’s healthy growth, viability/vitality and fermentation efficiency (Hill & Stewart, 2019). Though FAN strongly depends on malting conditions (Briggs, 1998), some FAN components (alongside reducing sugars) during mashing might provide minor flavour precursors that undergo Maillard reaction (Hill & Stewart, 2019; Hughes, 2009). Despite FAN influencing other fermentation factors (like cell biomass, growth, pH, viability, and attenuation rate) (Shimizu et al., 2002), too high FAN is undesirable given the resultant excessive yeast growth, which could affect beer stability (BYO, 2019b). Malts with higher FAN levels require adjuncts, which can act as nitrogen diluent but would contribute little-to-no TSN to the wort (Briggs et al., 2004).

Resembling (p > 0.05) across rice varieties (FARO 57 = 40 g/100 g > FARO 44 = 39 g/100 g > NERICA 7 = 37 g/100 g), the peak DE of rice wort (Refer to Table 7) suggested increased hydrolysis. Slight DE variations in wort might reflect the differences in the amylose-amylopectin ratio of rice starch. This is because the amylose can be more completely hydrolyzed than amylopectin, as the latter is limited by beta-limit dextrin due to branched chains (Osuji & Anih, 2011). Varietal differences, varying malting conditions as well as amount/type of enzymes developed in the rice grain particularly during malting might also govern the degree of hydrolysis and hydrolysates types obtained. The maltose and maltotriose remain the most abundant sugars present in malt/wort (Goldhammer, 2008; Palmer, 2009), which could also influence the DE of wort (Ofoedu et al., 2020).

Resembling (p > 0.05) across rice varieties (FARO 57 = 12.66 g/100 g > FARO 44 = 11.15 g/100 g > NERICA 7 = 9.68 g/100 g), OE of rice malt wort (Refer to Table 7), neared those of millet (~10 g/100 g), sorghum (10.42 g/100 g) and barley (11.0 g/100 g) malts (Agu, 1995), and compared well with other reported ranges (7.5–9, 8–9.5, 11–14 and 12.5–16 g/100 g) of different barleys used for ale beers (Papazian, 2006). Principally, original gravity (density) of the wort is four times the OE by Plato scale. Well-known, water density is 1.0000 at standard temperature and pressure (STP); if respective wort density of FARO 44, FARO 57, and NERICA 7 were 1.04448 (11.15 g/100 g), 1.05064 (12.66 g/100 g) and 1.03872 (9.68 g/100 g), the corresponding wort will be 44.48°, 50.64° and 38.72° of excess gravity. Thus, wort densities would consider the solution factors/mixtures of dissolved carbohydrate materials, soluble proteins and minerals that typically emerge from malted cereal materials. Though grain mashing considerably influences OE of wort, most grain modified products (that is, cell wall degradation and enzymatic breakdown) in endosperm’s protein-starch matrix (Agu & Palmer, 2001) would be released (into the wort) as soluble extracts.

Discussion of rice malt beer

Resembling (p > 0.05) across rice varieties (FARO 57 = 3.90 > FARO 44 and NERICA 7 = 3.80), pH of beer (Refer to Table 8) appeared lower than those of barley (4.1–4.5) (Bamforth, 2001) as well as sorghum (3.90–4.10) beers (Iwouno, Ofoedu & Ofoedum, 2019). Low pH in rice malt beer might be due to organic (weak) acids excreted by yeast with excess CO2 (which provides relative amounts of carbonic acid) during fermentation. Low beer pH also depicts its sharpness of taste. When pH is below 4, taste further sharpens with increased foam stability and head retention (Bamforth, 2006). By decreasing buffering capacity, lower pH increases yeast growth, removes colloidal particles of proteins-polyphenol complexes (and other insoluble materials) and inhibits microbial growth (in beer/wort) (Leiper & Miedl, 2006). In addition, pH in beer, determined by organic acids, for example acetic, lactic, pyruvic, and citric acid, can influence its flavour.

The depiction of beer colour is largely based on appearance, which remains critical to (product) acceptance. Importantly, this has largely been among the first quality attribute beer consumers perceive (Leon et al., 2006; Osuji et al., 2020). Beer colour resembled (p > 0.05) across rice varieties (FARO 57 = 3.73 °EBC > FARO 44 = 3.70 °EBC > NERICA 7 = 3.20 °EBC) (Refer to Table 8). Although fairly above those of another rice malt beer (1.70–2.60 °EBC) (Mayer et al., 2014), the rice malt beer herein compared well with those of typical barley (2.00–4.00 °EBC) (O’Rourke, 2002a), but not so for sorghum (6.0–6.6 °EBC), barley double crown (~7.5 °EBC) and barley rex (~14.0 °EBC) malt lager beers (Olatunji et al., 1993). It is largely understood that as °EBC increases, the beer colour gets darker. When assessed by the Saveur Bierre colour chart (Anon, 2020), the range of rice malt beer colour of the current study was perceived as pale yellow lager. Beer colour variations could be as a result of either decolourization of the (beer colour) substance as pH dropped (Kunze, 2004), changes/differences in malt colour, or inconsistencies in the colour formation of wort during boiling process (Briggs et al., 2004). Phenols (tannins) are natural organic compounds in malts/hops, which change beer colour from pale yellow to dark brown via Maillard reaction/caramelization (Whistler & Bemiller, 2008; Panthare, Opara & Al-Said, 2013). In addition, Maillard reaction and caramelization occurring independently/simultaneously would influence colour formation/intensity (Kunze, 2004). Other factors like pH level, yeast strain, hop usage, maturation duration, and specialty ingredients can influence beer colour.

Apparent extract resembled across rice varieties (p > 0.05) (NERICA 7 = 4.93 g/100 g > FARO 44 = 4.59 g/100 g > FARO 57 = 4.57 g/100 g) (Refer to Table 8). Noticeably, there appears some reduction in gravity of wort from 9.68 to 12.66 g/100 g (Table 4) to 4.57–4.93 g/100 g (Table 5) in the final rice malt beer. Dissolved solids (sugars, amino acids, minerals, among others) in wort utilized by yeast during fermentation might reduce the final beer gravity. As yeast utilizes sugars (and other compounds) to produce alcohol, the gravity of wort may well decrease (Boulton, 1991; Briggs et al., 2004). Moreover, fermentability of wort depicts the proportion of dissolved solids (extract) that can be fermented. In other words, 59% (FARO 44), 64% (FARO 57), and 49% (NERICA 7) of fermentable materials in these worts utilized by yeast produced AE of 4.59 g/100 g, 4.57 g/100 g and 4.93 g/100 g, respectively.

Alcohol content of beer, although differing significantly (p < 0.05) across rice varieties (FARO 57 = 4.13%ABV > FARO 44 = 3.54%ABV > NERICA 7 = 2.82%ABV) (Refer to Table 8), fell within a generally anticipated range (4–6%ABV) (Polan, Eisner & Vytras, 2015), somewhat above 2.55, 3.09, and 3.65%ABV of millet, sorghum, and barley beers, respectively (Agu, 1995). FARO 57 with peak fermentability of 64% corresponded to 4.13%ABV and NERICA 7 with least fermentability of 49% corresponded to 2.82%ABV. This suggests that alcohol concentration in beer does not solely depend on the OE/gravity of wort, but more likely on the availability of fermentable extracts, readily utilized by the yeast. Whilst the fermentable extracts especially sugars in wort remain the beer quality index (Jordao, Vilela & Cosme, 2015), its concentration (and subsequent utilisation) in the wort can help to determine the improved fermentation efficiencies (Zhao et al., 2008).

Resembling (p > 0.05) across rice varieties (FARO 44 = 5.30 NTU > FARO 57 = 4.80 NTU > NERICA 7 = 4.30 NTU), beer turbidity (Refer to Table 8) were above those of sorghum (1.6–2.0 NTU) and barley malt (3.2 NTU) lager beer (Olatunji et al., 1993), but below those of sorghum (red) (South Africa) (~12.8 NTU), sorghum (white) (Australia) (~28 NTU), sorghum (white) (Nigeria) (~33.2 NTU) malt beers (Aisen & Muts, 1987), and sorghum beer clarified with different filter aids (8.28–26.56 NTU) (Iwouno et al., 2019). Considering 1.00 EBC equals 4.00 NTU, the beer turbidity can be graded based on degree of haziness, which includes; brilliant: 0–0.50 EBC (0–2.00 NTU); almost brilliant: 0.50–1.00 EBC (2.00–4.00 NTU); very slightly hazy: 1.00–2.00 EBC (4.00–8.00 NTU); slightly hazy: 2.00–4.00 EBC (8.00–16.00 NTU); hazy: 4.00–8.00 EBC (16.00–32.00 NTU) and very hazy: > 8.00 EBC (>32.00 NTU) (Callemien & Collin, 2009). Herein, the rice malt beer (4.30–5.30 NTU) would be considered as ‘very slightly hazy’ (Refer to Table 8). Some proteins not removed during wort boiling, surviving fermentation, and finding its way into the beer, might equally cause the haze (Briggs et al., 2004). Besides the origin of haze formation as either biological (e.g. bacteria, cell debris, yeast) or non-biological (inorganic, carbohydrate-based and protein-polyphenol complexes) (Siebert, Carrasco & Lynn, 1996; Stewart, 2004; Briggs et al., 2004), beer haziness might be due to ineffective filtration, non-flocculant yeast, and or poorly modified malt/filter aids (Steiner, Becker & Gastl, 2010). Coloured compounds such as melanoidins (Iwouno et al., 2019), cereal/malt-type, and differences in chemical composition/processing methods can influence beer turbidity. In addition, centrifugation and microfiltration used during commercial production can also increase beer clarity (Kuiper et al., 2002; Shotripuk et al., 2005).

The sensory attributes (colour, aroma, taste, mouthfeel, appearance, and overall acceptability) of rice malt beers compared well with commercial lager beer (Refer to Table 9). Based on the hedonic scale, the panelists viewed the colour of the rice malt beers (6.66–6.91) as pale yellow colour and compared to the commercial lager beer (8.71). The panelists considered the rice malt beers as slightly liked compared to the commercial lager beer that was liked very much. The colour variations in the beer samples may be due to differences in kilning temperatures and chemical compositions (sugars and amino acids) that facilitate the formation of melanoidin in beer (Osuji et al., 2020; Iwouno et al., 2019). The panelists obtained the mouthfeel of rice malt beers (6.57–6.96) as slightly liked/relatively flat compared to the commercial lager beer (8.51) which was liked very much. The variations in the mouthfeel of beer samples may be due to varying concentrations of residual sugars, higher alcohols as well as organic acids in the beer (He et al., 2014; Iwouno, Ofoedu & Aniche, 2019).

We opine that the appearance of rice malt beer particularly from the consumer’s quality perspective, which itself could also include but not limited to the absence as well as colour of haze, would greatly affect beer perception. Similar to the mouthfeel and colour of beer samples, the appearance (6.24–6.52) of rice malt beer samples was slightly liked probably because the rice malt beers appeared very slightly hazy compared to the commercial lager beer (8.61) which appeared almost brilliant in clarity (Refer to Table 9). The variations in appearance could be due to differences in brewing technology adopted. Notably, the taste and aroma across the beer samples resembled (p > 0.05), although the sensory scores indicated the taste and aroma of commercial lager beer as liked very much, whereas that of rice malt beer samples were liked moderately (Refer to Table 9). Specifically, the aroma and taste of beer are characterized by volatile compound profile (Marconi et al., 2017) influenced principally by yeast metabolism. The differences in taste and aroma of beer samples may occur with fermentation by-products, such as aroma-active esters, higher alcohols, and aldehydes (He et al., 2014; Ferreira & Guido, 2018). The overall acceptance herein suggests sensory properties of beer might be affecting consumer liking, considering the commercial lager beer was liked very much by the panelists. Besides FARO 44 and NERICA 7 beer being liked moderately, the FARO 57 beer was slightly liked. Overall, the sensory profile of rice malt beer resembled that of commercial lager beer in aroma and taste, but flatter in mouthfeel (Refer to Table 9).

Conclusions

The characteristic changes in malt, wort, and beer from different locally produced (Nigeria) rice varieties as influenced by varying malting conditions were investigated. The rice varieties exhibited desirable gain quality characteristics and showed acceptable aptitude to be malted due to their germinative property of greater than 85%. Malting conditions significantly influenced the CWE, HWE, DP, MC, and TN of rice malt. Across varieties, the pH, TSN, Brix, KI, FAN, DE, and OE in rice malt wort, and pH, colour, AE, and turbidity in rice beer resembled (p > 0.05), but not so in %ABV (p < 0.05). In addition, the rice malt beer, very slightly hazy, represented a pale yellow light lager. To obtain wort that makes an alcohol clear-beer, requires the addition of exogenous enzymes, particularly in the mashing of rice malts. Moreover, malting improves hydrolysis, modifies the starchy (rice) endosperm, and allows adequate production of FAN, TSN, and other fermentable extracts in the wort. Besides the sensory profile differing in appearance, the characteristic pale yellow rice malt beer resembled the commercial lager beer in aroma and taste, but more flat in mouthfeel. Overall acceptance suggest that rice malt beer from FARO 44 was preferred more amongst other rice malt beers, after the commercial lager beer.

Although from the sensory observations, we see that the rice malt beer would differ from the commercial lager beer, we still believe the rice malt beer stands a chance to provide its own eccentric beer style. In addition to increasing the DP of rice malt (which has been demonstrated as dependent on both (rice) variety and varying malting conditions), varying malting conditions with respect to (rice) variety could play a vital role in reducing the cost of exogenous enzymes, particularly if the aim is to actualize an all-rice gluten-free beer. Another aspect of this study that we consider very important is the use of blends of rice malt as specialty ingredient (and not as an adjunct) together with barley malts in mashing/brewing, which has the potential to help save the cost of barley malt imports, as well as reduce the singular dependency on barley (temperate crop) for tropical brewing.

Overall, the malting conditions of the current study shows high promise for commercial lager beer production. Further research is warranted on other locally available rice varieties as well as underutilized cereals for malting/brewing, which would target higher extract yield as well as a clearer beer. However, there is a chance that not all the local rice varieties (in Nigeria as would be the case for future studies) would be suitable for brewing. Therefore, a careful and thorough variety selection would be needful if an optimised malt beer output is to be actualised. Another research direction of future studies could determine the foam formation, retention, bubble size, and distribution as well as microbiological analysis of rice malt beer from selected malting conditions. It is also recommended that future studies could quantitatively and qualitatively determine and characterize enzymes in rice malts of different varieties affected by varying malting conditions.

Supplemental Information

Supplemental Information 1 Raw Data.

Click here for additional data file.

Additional Information and Declarations

Competing Interests

Author Contributions

Data Availability

Charles Odilichukwu R. Okpala is an Academic Editor of PeerJ.

Chigozie E. Ofoedu conceived and designed the experiments, performed the experiments, analyzed the data, authored or reviewed drafts of the paper, and approved the final draft.

Chibugo Q. Akosim conceived and designed the experiments, performed the experiments, prepared figures and/or tables, and approved the final draft.

Jude O. Iwouno conceived and designed the experiments, prepared figures and/or tables, and approved the final draft.

Chioma D. Obi performed the experiments, prepared figures and/or tables, and approved the final draft.

Ivan Shorstkii analyzed the data, authored or reviewed drafts of the paper, and approved the final draft.

Charles Odilichukwu R. Okpala analyzed the data, authored or reviewed drafts of the paper, and approved the final draft.

The following information was supplied regarding data availability:

Raw data are available as a Supplemental File.

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
