# Peer review of "Characteristic changes in malt, wort, and beer produced from different Nigerian rice varieties as influenced by varying malting conditions"

_PeerJ, doi:10.7717/peerj.10968_

## Round 0.1 · original submission · Major Revisions

Please address the reviewers' comments carefully.

Reviewer 1 ·

Basic reporting

Overall quality: The subject of the paper is interesting and it fits well with the scope of PeerJ journal, and with conclusions of interest for the scientific community. Regarding the conception of the paper and the presented results, the paper can be graded as good, but requires some improvements and correction before publishing.
• The professional English language is used throughout the manuscript.
• The topic of the study is interesting, it fits into trends in science as well as in the manufacturing practice. Submitted paper is compliant with the aims and scope of the scientific journal PeerJ.
• The title cover the main aspects in this paper, it reflect the aim and scientific purpose of conducted experiment.
• Presented abstract doesn’t explain the significance of the paper. This part of the manuscript seem to be descriptive.
o Please rewrite it clearly stating the facts; focus more on how your research has contributed to knowledge gaps; describe research limitations for future research and restate your major findings.
o Instead of indicatively stating the obtained results, it would be better to explain the reasons for choosing the process parameters and substantiate them with the obtained results. Eg., for samples A, the application of the following production conditions is recommended in order to obtain more desirable quality parameters, which is confirmed by the performed sensory analysis.
• Introduction provides a good, generalized background of the topic giving the reader an appreciation of the wide range of applications for this technology.
• The methods used in this paper are appropriate to the aim of the study. Methods are clear and replicable.
• Results are presented in an appropriate format (clear and legible), well labelled and described. Raw data were supplied as supplementary file.
• The literature cited in this paper is relevant to the study. The paper use appropriate references in the correct style to promote understanding of the content.

Experimental design

The methods used in this manuscript are standard and appropriate to the aim of the study. Methods are clear and replicable. In this research, no measurement of parameters such as foam stability or bubble size and distribution, microbiological analysis of beer, etc. was performed. These parameters could greatly contribute to the quality of research.
The disadvantage of this paper is that it does not clearly highlight the scientific contribution that the results will bring. It could be said that the research question is not well defined (in terms of defining process conditions of beer production) /Line106-108.

Validity of the findings

Conclusions presented in this paper correlate to the results found. Given the scope of the results presented, it is necessary to improve conclusion section. Please rewrite it and focus more on how your research has contributed to knowledge gaps; add scientific and practical significance of the selected method.

• Uniform the numbering of the titles and subheadings of the paragraphs.
Line 34: Please check the expression “thousand corn weight”!
Line 40, 47: Correct ABV to %ABV.

M&M
• In general, the font and the way of writing mathematical equations should be standardized. Some mathematical equation also lack explanations of variables and/or units of measurement.
o Equations 1, 2, 7, 8 gives the full names of the variables, in Eq. 3, 4 and 5 is a mix, and in Eq. 6 and 9 only the short ones. Furthermore, in Equation 10, in addition to the full name of the variable, its abbreviation is also given. In some equations (eg Eq. 11) the unit of measurement is given next to the variable, while in others it is not. Align it.
• In some places in the text the term “rice malt” is used while in others it is just “malt”, this needs to be standardized.
• Where possible, add the number of iterations/determinations of the analysis performed.
Line 171: The title says a thousand grains of corn. Is that a mistake? Should “corn” be replaced by “grain of rice”?
Line 182: It is necessary to explain the variable "Dormancy" in the mathematical equation shown.
Line 191: What is the variable called X. State its units of measurement.
Line 192-195: Please state the units of measurement for the following variables: Wi, Wf, MC, D. There is missing explanation of the variable D.
Line 204: Delete the colon in the paragraph title.
Line 210: Explain the S.G variable. If the authors meant the variable Specific Gravity Degrees, then it should be clarified.
Line 211: Why the title of the paragraph begins with the number 2.4.1.2., when the previous paragraph is numbered 2.5.1.1?
Line 213: Should be written Lintner Degrees per Kilogram (°Lintner/kg).
Line 224-225: If it is a Lintner degree please correct “°L” to “°Lintner”. Lintner is an index that measures a malt’s diastatic power (DP), commonly written as “°Lintner”. To avoid possible misinterpretation of the results (degrees Lovibond or Lintner), it is recommended to write the DP unit of measurement as °Lintner. Please make mentioned correction in the Eq. 5.
Line 269: Add an explanation of the variables TSN and TN.
Line 319: What about measurements performed in triplicates (Line175)?

R&D
Line 538 and 651: Change ABV to %ABV.
Table 3, 4: To avoid possible misinterpretation of the results (degrees Lovibond or Lintner), it is recommended to write the Lintner degree as “°Lintner”. Please apply this change to all Discussion section.

·

Basic reporting

The English of the current manuscript is vey hard to understand sometimes.
the author has not cited most relevant references and so the manuscript is lack of novelty. The knowledge that the author want to display is not new for the brewing industry.
There was also no hypotheses within the current manuscript.

Line 76-77: why suddenly jumped from steeping to mashing? how about germination?
Line 78: how about other sugar?
Line 82-84, 85-87, what did the author mean here?
Line 83-84, why the author say that protein degradation is responsible for the cell wall degradation?
Line 88-89, why? in what aspects?
Line 94-95, any references?
Line 97-98, this is what you have done, not the research goal of the current study.

Experimental design

The experiment was not well-designed. For example, the author has not mentioned why rice is a promising substitute for barley, as rice is the staple foods for human? in Line 80, this is not the reason that why the content of starch in rice is higher than that of barley. There are must be other reasons.
The mashing procedure was really really complicated and made no sense. This is definitely not the proper mashing procedure for the brewing industry.

Line 117, what alpha-amylase and beta-amylase? from which company? how about the enzyme activity?
Line 118, again, what protease?
Line 145, what is enzyme activity?
Line 153, why monitor starch?
Line157-158, the English is really bad and the author has no cited any relevant references.
Line 329-330, any references?
Line 334-335, why did not measure the starch content?

Validity of the findings

the findings of the current study was not new at all and there are already many many studies concerning the subject.

Reviewer 3 ·

Basic reporting

The authors conducted a extensive research about rice malt modifications in different malting conditions and its influence on wort and beer. The data set is extensive and intesting to scientific audience dealing with brewing and malting. I agree that the article meets the standards of the journal.

Experimental design

The experiment is well designed and extensive. Perhaps the authors could reduce the number of pages by combining tables, and reducing the M&M section.

Validity of the findings

The finding are valid and all explenations have been provided in the R&D part.

Additional comments

The paper is too long and should be significantly reduced prior acceptance (examples of how to reduce it are stated in the text before).

---

## Round 0.2 · accepted · Accept

The authors have revised the manuscript addressing the reviewers' comments.

Reviewer 1 ·

Basic reporting

The topic of the paper is interesting and it fits well within the framework of the journal PeerJ and conclusions that are of interest to the scientific community. In terms of the conception of the paper and the results presented, the paper can be considered good. The revisions that the authors have made to the manuscript are very effective.

Experimental design

The revisions that the authors have made to the manuscript are very effective. The research question is well defined, relevant and meaningful. The methods are described in sufficient detail with notes on replication.

Validity of the findings

The authors made many corrections that improved the quality of the paper. The results presented in the manuscript are interesting. The conclusions are well stated, linked to the original research question, and limited to supporting findings. The conclusions presented in the paper correlate with the results found.

Additional comments

The authors have incorporated all reviewer comments into the revised version of the manuscript. I am satisfied with their responses and corrections. Therefore, I recommend the publication of the manuscript as it is.

Reviewer 3 ·

Basic reporting

The paper is extensive and I feel that the authors should reduce the number of pages or divide it to two papers.However, wheather this should be reduced or not, I would leave the decision to the Editor. Also, please add rice into the title, since the title does not give a clear insight into the whole subject matter.

Experimental design

The experimental design OK.

Validity of the findings

OK.

Additional comments

You did an extensive research and obtained results that could be useful for future research and to scholars as guidlines for rice malting.